# Ultra-Narrow Bandwidth Microwave Photonic Filter Implemented by Single Longitudinal Mode Parity Time Symmetry Brillouin Fiber Laser

**DOI:** 10.3390/mi14071322

**Published:** 2023-06-27

**Authors:** Jiaxin Hou, Yajun You, Yuan Liu, Kai Jiang, Xuefeng Han, Wenjun He, Wenping Geng, Yi Liu, Xiujian Chou

**Affiliations:** 1Key Laboratory of Instrumentation Science and Dynamic Measurement Ministry of Education, North University of China, Taiyuan 030051, China; jiaxin_hou@163.com (J.H.); 13100101409@163.com (Y.L.); jk19980923@163.com (K.J.); xuefenghan1998@163.com (X.H.); hewenjun@nuc.edu.cn (W.H.); chouxiujian@nuc.edu.cn (X.C.); 2School of Aerospace Engineering, North University of China, Taiyuan 030051, China; 3School of Semiconductor and Physics, North University of China, Taiyuan 030051, China; wenpinggeng@nuc.edu.cn

**Keywords:** microwave photonic filter, single longitudinal mode Brillouin laser, parity-time symmetry, ultra-narrow bandwidth, large tuning range

## Abstract

In this paper, a novel microwave photonic filter (MPF) based on a single longitudinal mode Brillouin laser achieved by parity time (PT) symmetry mode selection is proposed, and its unparalleled ultra-narrow bandwidth as low as to sub-kHz together with simple and agile tuning performance is experimentally verified. The Brillouin fiber laser ring resonator is cascaded with a PT symmetric system to achieve this MPF. Wherein, the Brillouin laser resonator is excited by a 5 km single mode fiber to generate Brillouin gain, and the PT symmetric system is configured with Polarization Beam Splitter (PBS) and polarization controller (PC) to achieve PT symmetry. Thanks to the significant enhancement of the gain difference between the main mode and the edge mode when the polarization state PT symmetry system breaks, a single mode oscillating Brillouin laser is generated. Through the selective amplification of sideband modulated signals by ultra-narrow linewidth Brillouin single mode laser gain, the MPF with ultra-narrow single passband performance is obtained. By simply tuning the central wavelength of the stimulated Brillouin scattering (SBS) pumped laser to adjust the Brillouin oscillation frequency, the gain position of the Brillouin laser can be shifted, thereby achieving flexible tunability. The experimental results indicate that the MPF proposed in this paper achieves a single pass band narrow to 72 Hz and the side mode rejection ratio of more than 18 dB, with a center frequency tuning range of 0–20 GHz in the testing range of vector network analysis, which means that the MPF possesses ultra high spectral resolution and enormous potential application value in the domain of ultra fine microwave spectrum filtering such as radar imaging and electronic countermeasures.

## 1. Introduction

A microwave photonic filter (MPF), as the core component of microwave photonic signal processing systems, is key technology to address the bottlenecks faced by next-generation wireless RF systems in terms of speed, bandwidth, volume, and power consumption [1]. As a key functional component in the front-end of information system transmission and reception to achieve target signal screening and interference signal suppression, the passband width of the filter directly determines the performance of frequency selection. With the demand for kHz or even Hz filtering in fields such as filtering strong interference signals [2], high-precision radar system imaging [3], and high sensitivity microwave photon sensing [4], the development of ultra narrow band microwave photon filters has attracted increasing attention in recent years in overcoming electronic bottlenecks, generating, transmitting, and processing RF signals. In recent years, many integrated microwave photonic chips have been reported in different technical fields [5,6,7]. This is an important breakthrough to prove the feasibility of integrated programmable microwave photonic processors. And these research results also provide a new way for wireless communication and defense applications.

The importance of microwave photonic filters in overcoming electronic bottlenecks and improving the performance of information systems in generating, transmitting, and processing radio frequency (RF) signals is axiomatic. Due to the ultra-high frequency characteristics of light waves, achieving an ultra-narrow filtering bandwidth and a wide tuning range has always been the goal pursued in the research of MPF. The ultra narrow filtering bandwidth means that MPF has high frequency selection characteristics, while the wide tunable range indicates that MPF has the ability to dynamically tune filtering. In fact, the essence of MPF is to map optical filters to microwave filters, and the filtering performance largely depends on the quality of signal processing of the optical RF signal in the optical domain. Therefore, in order to achieve ultra-narrow bandwidth microwave signal processing, high-precision optical signal processors are actually required. So far, various schemes and strategies have been proposed to achieve narrowband tunable filtering, such as Fabry-Pérot(FP) cavity [8], fiber Bragg grating [9], cascaded microring resonator [10], Sagnac loops [11], Mach Zehnder interferometer [12,13] and the SBS effect [14]. Among them, Brillouin MPF (based on SBS) as a new type of coherent type filter that achieves optical signal processing based on the interaction between light and matter in optical devices has been a hot research topic in recent years. It exhibits excellent processing capabilities in the field of narrowband signal processing due to the inherent high gain, narrow linewidth, and low threshold characteristics of SBS. It has been one of the best choices for achieving high-resolution and high suppression ratio MPFs. The basic principle of MPF operation mentioned above is to use the combination of Brillouin Stokes gain and optical modulated sideband signal, and achieve efficient frequency selection filtering of MPF by the phase intensity modulation (PM-IM) conversion of SBS [15,16,17]. However, There are still bottlenecks in the design of ultra narrow band MPF, such as fixed Brillouin gain spectral line width and difficulty in further compression. Due to the limitation of the material phonon lifetime [18], the Brillouin gain spectral linewidth of most material media is mostly in the MHz level. Using SBS gain to directly process the optical RF signal, its corresponding passband response is still at the MHz level. It is difficult to satisfy the filtering demands of certain special situations (such as microwave photon sensing, electronic interference, etc.), where the useful RF signal is only distributed within the ultra-narrow bandwidth of kHz order or below. Based on these research foundations, various methods for improving filter bandwidth and compressing it to a narrower range have been reported in many literature [19,20,21,22,23] in recent years. Zhang et al. [20] achieved an MPF which has a bandwidth of 7.8 MHz and a gain response of 24 dB by clipping one Brillouin gain spectra with two Brillouin loss spectra. However, to pay for the loss caused by maximum gain, this program has higher requirements for pump power. Duan et al. [21] achieved around 300 kHz ultra narrow linewidth based on SBS and an optical-electrical feedback loop (OEFL), but this method lacks structural simplicity relatively. Xu et al. [23] presented a 114 Hz bandwidth MPF, and the stimulated Brillouin scattering gain spectrum is narrowed by the double-ring Brillouin fiber laser (DR-BFL) with the Vernier effect. Inspired by this method, in the design of MPF, we can use a Brillouin laser resonator, which does not require additional fiber ring resonators, but more importantly, the ultra-narrow filtering bandwidth can be achieved by directly taking advantage of the inherent narrow linewidth performances of Brillouin lasers.

As the aspect of solving the tuning problem, there are various wide tunable range methods that have been proposed to enhance the flexibility of MPF [17,24,25,26,27,28,29,30,31]. Essentially, achieving tunability is to change the central frequency or free spectral range of the filter by changing its structural parameters or external physical parameters. Therefore, the above methods can be summarized into two main categories. One way is to adjust the central frequency of MPF by tuning the travel time of signal stresssing light in a dispersed media. It is natural to alter the size of dispersion medium. For example, the passband center frequency of the proposed MPF is able to be continuously adjusted by adjusting the length of the dispersion fiber delay line [25]. But the tuning method of this MPF is full of complexity and hard to achieve continuous tuning. The other way is to tune the filter passband of MPF through regulating the wavelength of the SBS pump light. At present, there have been many related studies, such as Gao et al. [27] proposing a tunable dual passband MPF based on PM-IM conversion, which uses a phase modulator and an equivalent phase shifted fiber Bragg grating (EPS-FBG). The central frequencies of the passbands can be tuned by shifting the wavelengths of the pump light. Similarly, Yuan et al. [30] and Xu et al. [23] proposed that by directly adjusting the wavelength of the pump used to excite Brillouin gain, to achieve the tunable MPF. The research results indicate this method is convenient and efficient, and can achieve uninterrupted large-scale tuning of MPF.

Based on the above analysis, we can see that the MPF technology based on lasers is considered to be one of the most promising means for achieving narrowband filtering and wide tunability [22,23,32,33,34]. Wherein, a MPF [32] using a multiwavelength erbium-doped fiber laser (EDFL) is demonstrated, but its central frequency is not tunable. A reconfigurable and continuously tunable MPF by applying a Mach–Zehnder interferometer (MZI) and a controllable multi-wavelength fiber laser based on Raman fiber gain is presented in [33]. However, the MZI in this device is expensive, bulky, and susceptible to environmental interference. An approach for achieving single channel MPF utilizing polarization modulators and broadband tunable FP lasers has been presented in [34]; however, because the modulated signal carried by the signal light is obtained by VNA modulating the electro-optic modulator (EOM) with RF scanning signal, the performance of the EOM limits its adjustable range. Recently, in [22,23], the authors achieved a sub-kHz bandwidth MPF based on a double-ring Brillouin fiber laser. However, the vernier effect used in dual ring resonators has the disadvantage of difficulty in matching different free spectral range (FSR) peaks, which limits mode selection efficiency.

In recent years, the PT symmetric photonic system has attracted people’s attention and has been widely studied [35,36,37,38] due to its advantages such as releasing of FSR matching. Moreover, a series of lasers, like on-chip micro ring lasers [39,40,41] and fiber lasers [42,43], utilize the existence of PT symmetry and can be used for laser mode selection, which effectively solved the contradiction between low phase noise of output signal and single mode output. For example, in [41], a wavelength tunable micro ring laser was shown, realizing PT symmetry based on two electrically pumped micro ring resonators and generated a single frequency light wave with a side mode suppression ratio of 36 dB. In [43], the PT symmetric fiber laser is proved, where PT symmetry is achieved by a single physical circuit with two equivalent intercoupling circuits based on orthogonal polarization. Based on the above analysis, through the PT system, there is no need to configure optical narrowband filters or the precise matching problem of FSR required by the Vernier effect, and single longitudinal mode laser output can be achieved. If we combine the PT symmetry system with SBS effect and fully utilize the advantages of both, the ultra-narrow linewidth single mode Brillouin laser can be produced without FSR precise matching, mode competition and small mode spacing in the mode selection based on Brillouin resonator, thereupon then achieving MPF with ultra-narrow bandwidth sigle pass band tunable filtering.

Based on the above innovative ideas, we propose and experimentally verify an ultra-narrow bandwidth and tunable MPF with flexibility based on a PT symmetric Brillouin laser. Its core innovation is to combine the narrow linewidth of Brillouin fiber laser resonators with the principle of PT symmetric system mode selection, and propose a new idea of using a single polarization-dependent Sagnac ring for cascaded PT systems in single mode Brillouin lasers. By selectively amplifying the optical microwave signal through the Brillouin laser gain with an ultra-narrow linewidth, and suppressing the side mode through a PT symmetric system, the ultra-narrow single passband filtering response is achieved. Specifically, the Brillouin fiber ring resonator converts Brillouin pump light with wide bandwidth into multimode Brillouin laser with narrow bandwidth, and then through an optical circulator, and then uses a cascaded PT symmetric system (Sagnac loop composed of two PCs and one PBS) for mode selection. We tune the coupling coefficient of gain and loss of the PT resonant cavity by adjusting the polarization state characteristics of light to achieve PT symmetry breaking and effectively suppress laser edge modes, thereby obtaining a stable single ultra-narrow bandwidth single passband filtering response. The dynamic tunability of the filter passband of MPF has been achieved by tuning the wavelength of Brillouin pump light to shift the position of Brillouin laser gain.

## 2. Operation Principle and Methods

This paper achieves MPF with ultra-narrow bandwidth and wide tuning range based on a single longitudinal mode PT symmetric Brillouin fiber laser. The basic principle is to use the amplitude frequency and phase frequency characteristics of the SBS effect to change the amplitude and phase of the carrier and sideband of the optical modulation signal, thereby achieving filtering in the optical domain. By combining the Brillouin resonant cavity with a MPF and utilizing the significant gain spectral linewidth compression characteristics of the Brillouin resonant cavity, the filter passband is narrowed to the sub-kHz level. On this basis, the PT symmetric system is cascaded through an optical circulator, which realizes the Sagnac effect through a physical optical fiber ring, so that the optical path through which two polarized light beams pass in the system has the same geometric shape, maintaining the balance between gain and loss. The polarization state of the light on both sides of the injected PBS is adjusted through a PC, so that the gain and loss of the two equivalent circuits can be controlled. If the gain and loss coefficients are greater than the coupling coefficient, the PT symmetry is broken, and the currently selected mode starts to oscillate, realizing the side mode suppression of the filter passband. The ultra narrow filtering bandwidth is eventually acquire by beating the frequency. Meanwhile, the wavelength change of SBS pump light leads to a change in the position of Brillouin gain, achieving large-scale tuning of MPF.

The proposed MPF scheme for experimental demonstration is shown in Figure 1. Overall, the experimental device is able to be approximately divided into two parts, in which the upper part of the optical path is used as an optical carrier to carry modulated signals, and the lower branch is the signal processing part of the Brillouin laser resonator and PT symmetric system. The laser1 is emitted by NKT1(NKT X15, line width 100 Hz), and the laser2 is emitted by NKT2(NKT X15, line width 100 Hz). The upper branch is an optical carrier, and the lower branch is a pump light.

Figure 2 demonstrates the schematic of the filtering process principle of the proposed MPF. The RF sweep signal $f_{\mathrm{RF}}$ output by vector network analyzer (VNA) is applied to the phase modulator (PM, ixblue MPZ-LN-20) to achieve double sideband modulated (DSB) of the optical carrier, whose frequency is $f_{c1}$. Figure 2a displays the spectrum of DSB modulated signal. Then, the upper sideband $f_{c1}+f_{\mathrm{RF}}$ and the lower sideband $f_{c1}$ − $f_{\mathrm{RF}}$ are entered in the Brillouin laser resonator with optical carriers by the coupler OC1 (OC1, 50:50).

In the upper part of the optical path, the laser2 frequency is $f_{c2}$, which is boosted by the erbium-doped fiber amplifier (EDFA) through PC2. Then, after passing through circulator1 (Cir1), laser2 enters in the 5 km single mode fiber (SMF), becoming the pump light. When the Brillouin threshold is exceeded by the power of the pump light, the self-induced SBS in the 5 km SMF can be stably excited, which is achieved by changing the output power of EDFA. The pump light frequency is reduced to $f_{c2}$ − $f_{B}$, where $f_{B}$ represents the Brillouin frequency shift, and is approximately 10.737 GHz in the research as illustrated in Figure 2b. And the Brillouin gain spectrum is represented by $\Delta f_{B}$ in Figure 2b. In SMF, the SBS gain spectrum and the optical modulation signal exists in an interplay, and the upper sideband part of the optical modulation signal located within the Brillouin gain spectrum is amplified by the Brillouin gain to achieve PM-IM conversion, as represented in Figure 2c. And the Brillouin laser resonator (consists of Cir1, SMF, OC1 and OC3) is injected by the selectively amplified optical signals through Cir1 and forms periodic resonance in the resonator. The length of the resonant cavity determines the period of resonance, and its structure makes the resonant linewidth extremely narrow, as shown in Figure 2d, providing an excellent frequency selection mechanism for the Brillouin laser resonant cavity, ultimately obtaining a Brillouin gain spectrum with extremely narrow linewidth.

The Brillouin laser resonator is connected to the PT symmetric system through Cir2. Resonant light passes through Cir2 to the Sagnac loop, which is composed of Cir2, 50:50 OC2, PBS, and the PCs (PC3 and PC4). As the incident light splits and moves in clockwise (CW) and counterclockwise (CCW) directions, two intercoupled rings are formed; one is gain, while the other is loss. By regulating PC to change the polarization state of linearly polarized light pumped into PBS, precisely controlling of gain and loss of CW and CCW light wave corresponding loop is realized. Due to the propagation of two light waves in the same polarization dependent Sagnac loop, the lengths of the two matching loops are basically identical. As shown in the Figure 2e, if the gain and loss have a good match and exceed the coupling coefficient, the PT symmetry is disrupted, leading to mode selection. In order to achieve the tunable characteristics of MPF, the pump frequency can be shifted by adjusting the wavelength of NKT2, thereby achieving the shift of the center frequency of SBS gain spectrum.

Eventually, the coupler (OC3, 10:90) is injected with output light, and 90% of which is entered counterclockwise in the Brillouin laser resonator to keep on resonance oscillation, and 10% of which entered in OC4 (OC4, 50:50) is divided into two beams by OC5 (OC5, 50:50), one is monitored by the photodetector (PD, Finisar XPDV21), and the other is detected by the optical spectrum analyzer (OSA) with 0.01 nm resolution. The VNA displays the frequency response of the MPF proposed in this article.

## 3. Experiment Results and Discussion

Next, we conducted experimental verification based on the above theories. Firstly, before conducting filtering performance testing, we conducted preliminary tests on the spectra of the pump light injected into the 5 km SMF, the spectra of the Brillouin Stokes light of fiber ring resonater (without PT symmetric system), and the spectra of the mixed light of Brillouin Stokes and optical RF signals. The comparison results of spectral measurements are shown in Figure 3. We set the initial wavelength of the pump light (NKT2) to 1550.00 nm, as shown by the blue line in the spectral comparison diagram. By adjusting the amplification power of EDFA until the stokes spectrum appears on OSA. When the amplification power of EDFA is 21 dBm, the stokes spectrum output by the fiber ring oscillator is shown by the red line in Figure 3, with a wavelength of approximately 1550.08 nm. Obviously, it can be concluded that the difference between the pump light wavelength and the stokes wavelength is 0.08 nm, which is corresponding to a frequency shift of approximately 10 GHz in the frequency domain. This result is basically consistent with the magnitude of Brillouin frequency shift in standard single-mode fibers and indicates that SBS is excited inside the fiber at this time. When the PM modulates the RF sweep signal emitted by VNA to the optical carrier and injects it into the Brillouin resonator, the mixed light of the SBS spectrum and DSB modulated optical signal are displayed in black lines in Figure 3, which demonstrated dual wavelength characteristics on the spectrogram. The left beam is the optical carrier output spectrum of NKT1, and the Brillouin laser spectrum of the composite resonator (as shown in the light blue marked area in Figure 1, a ring resonator with a cascaded PT symmetric system) is on the right side of the beam.

### 3.1. The Filtering Results of MPF in This Work

In Figure 4, we comprehensively compare the MPF passband response of optical microwave signal processing using a Brillouin intrinsic gain, a Brillouin fiber laser resonator, and a Brillouin fiber laser resonator cascaded PT symmetric, respectively. When the lower branch in the optical path structure of Figure 1 does not form a ring-resonant cavity, the bandwidth of the obtained filtering passband is equal to the intrinsic Brillouin gain linewidth of silica, as shown by the black line in Figure 4. After building the ring-resonant cavity shown in the light blue area of Figure 1 without PT symmetry structure, the frequency response of the MPF is shown in the blue line of Figure 4. The periodic resonance of the resonant cavity is due to the multiple amplifications of Brillouin Stokes in the resonator, and its linewidth is much smaller than that of the Brillouin gain spectrum. And due to the resonant cavity with the circular structure, the periodic response forms a comb shaped spectrum, and the spacing between the comb teeth is equal to the FSR of the ring resonant cavity. The red line illustrates the frequency response of MPF based on PT symmetric Brillouin fiber laser. In a PT symmetric system based on polarization diversity implemented in a single spatial resonant cavity, by controlling the polarization characteristics of light, the gain and loss coefficients of the resonant cavity can be tuned to achieve PT symmetry breaking [43]. When PT symmetry is broken, the gain difference between the main mode and the edge mode is significantly enhanced, making single mode oscillation possible. Based on such reasons, the PT symmetrical system has a good effect on suppressing side mode of the Brillouin laser, and the side mode suppression ratio is more than 18 dB. Moreover, the narrowest −3 dB bandwidth of MPF is low to 72 Hz (sub-kHz), exhibiting peerless ultra-narrow passband response, as shown in Figure 5. This result reduces the filtering bandwidth by about $10^{6}$ times compared to the MHz level filtering through the Brillouin intrinsic gain spectrum (approximately 13.816 MHz in this article).

### 3.2. The Mode Selection Mechanism of PT Symmetric System

Figure 6 shows the mode selection mechanism of the PT symmetric system. Based on Figure 1, the light entering OC2 is divided into two beams and travels in both clockwise (CW) and counterclockwise (CCW) directions to form two mutually coupled loops. The gain and loss of this structure can be accurately regulate by adjusting PC3 and PC4. In [44], the coupling equation of the mode in the cavity is given by the following equation:(1)$$\frac{d}{dt}\left[\begin{array}{c} A_{n}\\ B_{n} \end{array}\right]=\left[\begin{array}{cc} -i\omega_{n}+g_{\mathrm{An}} & ik_{n}\\ ik_{n} & -i\omega_{n}+g_{\mathrm{Bn}} \end{array}\right]\left[\begin{array}{c} A_{n}\\ B_{n} \end{array}\right]$$

In Formula (1), $A_{n}$ and $B_{n}$ express the oscillation of the nth longitudinal mode in the gain and loss circuit of SBS, $\omega_{n}$ represents the intrinsic frequency of the *n*th mode without PT symmetry. $g_{\mathrm{An}}$ and $g_{\mathrm{Bn}}$ are the net gain and loss coefficients of the two coupled loops in the case of the nth mode, which are also changes with variation in SBS gain and inherent loss of the laser cavity. And $k_{n}$ represents the coupling coefficient between the two coupled circuits of the nth mode in a PT symmetric system.

By solving Equation (Equation 1), the intrinsic frequency of the PT symmetric system can be obtained, which is given by the following Equation (Equation 2):(2)$$\omega_{n}^{\left(1,2\right)}=\omega_{n}+i\frac{g_{\mathrm{An}}+g_{\mathrm{Bn}}}{2}\pm \sqrt{k_{n^{2}}-{\left(\frac{g_{\mathrm{An}}-g_{\mathrm{Bn}}}{2}\right)}^{2}}$$

Supposing that the single mode output of the shown mode is achieved by rotating PC3 and PC4, the precise PT symmetry condition can be met. We can obtain $g_{\mathrm{An}}=-g_{\mathrm{Bn}}=g_{n}$, and the Equation (Equation 2) can also be written as
(3)$$\omega_{n}^{\left(1,2\right)}=\omega_{n}\pm \sqrt{k_{n^{2}}-g_{n}^{2}}$$

If $g_{\mathrm{Bn}}\;$ < $\;g_{\mathrm{An}}$, the system is at a low-level gain/loss state and operates in a PT-symmetry uninterrupted state, which will result in frequency splitting of a mode. And if the gain/loss is greater than the coupling coefficient, $g_{\mathrm{Bn}}$ > $g_{\mathrm{An}}$, for a conjugate mode that experiences gain while the other experiences decay when the other modes remain neutral, the PT symmetry broken condition is satisfied, as seen in Figure 6b.

In addition, if the SBS gain of the primary mode $g_{0}$ (0th) exceeds the oscillation threshold, while the SBS gain of other modes is below the threshold, the Single Longitudinal Mode can also be achieved in conventional BFL. Under this condition, the maximum gain contrast of SBS is determined by
(4)$$g_{\max}=g_{0}-g_{1}$$
where $g_{1}$ represents the SBS gain coefficient of the second largest competitive mode. Since the difference between $g_{0}$ and $g_{1}$ is very low, achieving a stable single-mode laser is quite difficult.

In contrast, we obtained the gain difference based on Equation (Equation 3) under the PT symmetry setting, which is determined by
(5)$$g_{\max\_\mathrm{PT}}=\sqrt{g_{0}^{2}-g_{1}^{2}}$$

In order to improve the quantization mode selection, the gain enhancement we calculate, known as gain contrast too, is referred to as the gain difference ratio between a PT symmetric laser source and a traditional one, as shown in the following equation.
(6)$$G=\frac{g_{\max\_\mathrm{PT}}}{g_{\max}}=\sqrt{\frac{g_{0}/g_{1}+1}{g_{0}/g_{1}-1}}$$

Due to $g_{0}$ > $g_{1}$, the gain difference is significantly enhanced, and the stability of SLM laser in the primary mode is ensured. To sum up, utilizing the mode selection effect of the PT symmetric system, a single longitudinal mode Brillouin laser can be obtained without the need for expensive high Q-value optical filters or the complex FSR peak matching process of the Viener effect.

### 3.3. The Tuning Range and Stability of MPF

Figure 7 shows that the MPF responses at different center frequencies during the process of changing the laser wavelength (i.e., pump wavelength) of the lower branch in Figure 1 from 1549.9200 to 1550.0800 nm. The image shows that the central frequency tuning interval of the filter is approximately 2 GHz, which can be stably tuned from 0 Hz to 20 GHz. Obviously, the passband side mode of MPF based on PT symmetric Brillouin laser is significantly suppressed. Figure 8 shows the out of band suppression ratio and amplitude of MPF at various center frequencies, demonstrating the stability and ability to suppress sidebands of MPF. The side mode suppression ratio measured within the tuning range are 24.1, 20.6, 21.1, 23.3, 21.6, 22.0, 20.4, 20.4, 22.2, 22.3, and 23.4 dB, respectively, with fluctuations of less than 3.7 dB. The measured amplitude of the MPF passband based on PT symmetry Brillouin laser in the tuning range are −30.0, −32.0, −33.0, −30.0, −36.4, −36.0, −35.5, −39.1, −34.7, −36.6, and −31.6 dB, respectively, with fluctuations less than 9.1 dB. Two curves indicate that the MPF raised in this paper has higher frequency selectivity and good stability. The sensitivity of the SBS effect to changes in strain and temperature is the main reason for this fluctuation. In addition, the coupling efficiency between the Brillouin laser in the resonant cavity and the signal optical carrier can also cause peak fluctuations in the filtering passband. In the experiment, fluctuations caused by the above reasons can be reduced by controlling the ambient temperature and adjusting the resonant cavity PC.

Table 1 compares the proposed MPF based on PT symmetric BFL with various implementation methods reported in recent literature based on SBS, including a summary of basic parameters such as filtering bandwidth and tuning range. From the table, we can conclude that the proposed scheme to narrow the bandwidth of MPF so far is mainly distributed in the MHz or kHz range, while the scheme to narrow the bandwidth of MPF in our work, which has obvious advantages, is substantially within the sub-kHz range. The comparison of MPF performance shows that the filtering method proposed in this paper has significant advantages in achieving ultra narrow single pass band tunable microwave signal processing.

## 4. Conclusions

In conclusion, we propose and experimentally prove a tunable narrow-band MPF based on a PT symmetric Brillouin fiber laser with a Hz bandwidth level. The biggest innovation of this article lies in the realization of ultra narrow band microwave photon filtering based on a single longitudinal mode Brillouin laser using a cascaded PT symmetric system Brillouin resonant cavity, and there is no need to configure ultra narrow band optical filters and perform complex FSR matching during the Brillouin laser mode selection process, i.e., due to the significant gain spectral linewidth compression characteristics of Brillouin laser, the passband of the filter is narrowed. Meanwhile, the PT symmetry system composed by a Sagnac ring consisting of two PCs and one PBS is cascaded into the Brillouin fiber laser resonator through the optical circulator to realize the MPF. The polarization state of the light on both sides of the injected PBS is changed through the PC, and the gain and loss of the two equivalent circuits are regulated. When the gain and loss coefficient exceed the coupling coefficient, the PT symmetry is broken, and the side mode suppression of the filter passband is realized, the ultra-narrow filter passband is obtained. The filter passband of the proposed MPF can be adjusted by changing the excited Brillouin pump light wavelength. The above results indicate that the proposed MPF achieves a single pass band narrow to 72 Hz and the side mode rejection ratio of more than 18 dB, with a center frequency tuning range of 0–20 GHz, which means that the filter has ultra high spectral resolution and can be used for hyperfine spectrum filtering systems.

## Figures and Tables

**Figure 1 micromachines-14-01322-f001:**
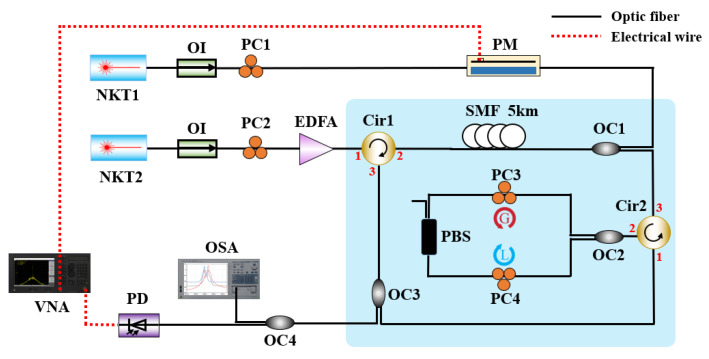
MPF experimental device. The light blue marked area represents the ring resonator with cascaded PT symmetric system. OC, optical coupler; OI, optical isolator; Cir, circulator; PM, phase modulator; PC, polarization controller; PD, photodetector; EDFA, erbium-doped fiber amplifier; SMF, single mode fiber; PBS, polarization beam splitter; VNA, vector net analyzer; NKT, the narrow linewidth laser; OSA, optical spectrum analyzer.

**Figure 2 micromachines-14-01322-f002:**
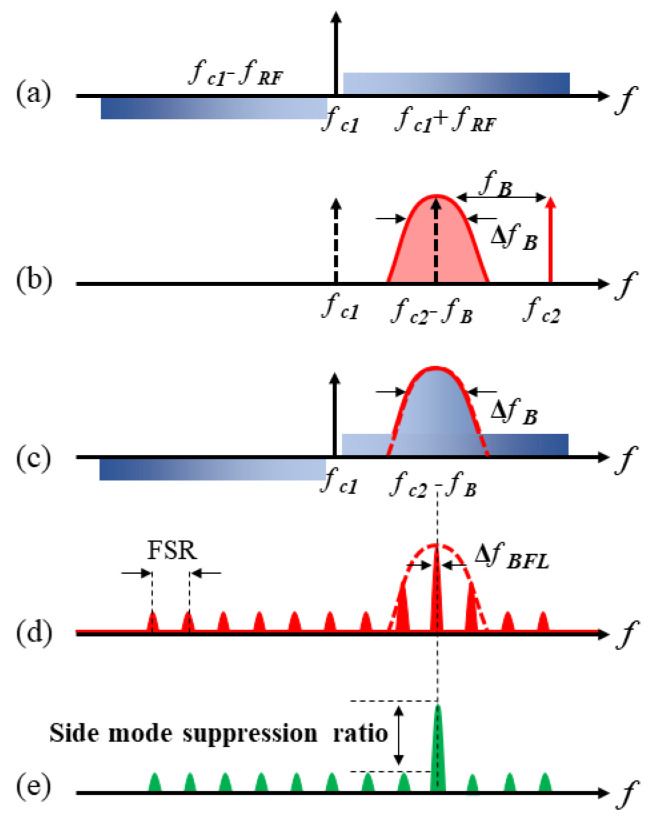
MPF operating principle. (**a**) Spectral schematic diagram of the DSB modulated signal (**b**) Spectral schematic diagram of the SBS. (**c**) Schematic diagram of SBS amplifying the upper sideband of DSB modulation signal. (**d**) Schematic diagram of FSR response. (**e**) Schematic diagram of MPF response.

**Figure 3 micromachines-14-01322-f003:**
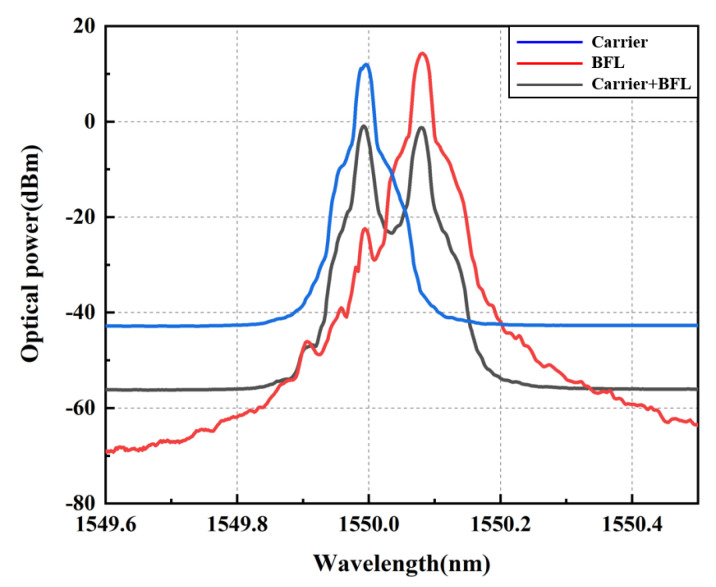
Spectrogram of pump light entering SMF, spectrum of SBS frequency-downshifted in the 5 km SMF, and the combined spectrogram of SBS and DSB modulated signal.

**Figure 4 micromachines-14-01322-f004:**
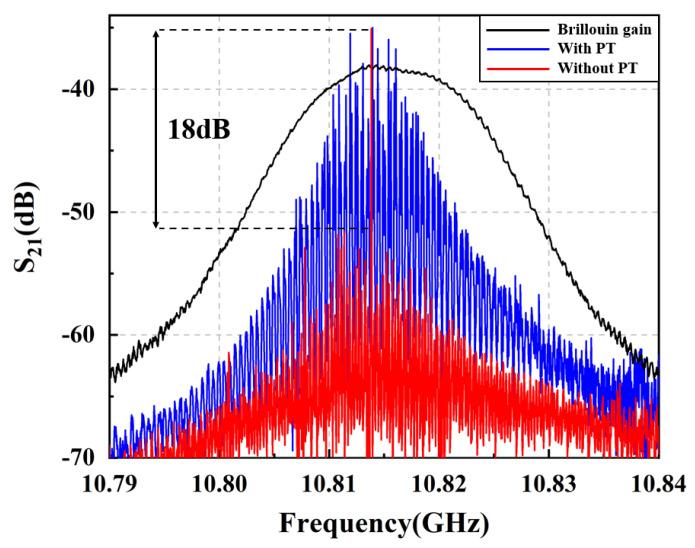
Corresponding to the frequency response of three various MPF structures. The black line represents optical microwave signal processing using a Brillouin intrinsic gain; the blue line represents a single ring Brillouin laser resonator; the red line represents a cascaded PT symmetric structure Brillouin laser resonator.

**Figure 5 micromachines-14-01322-f005:**
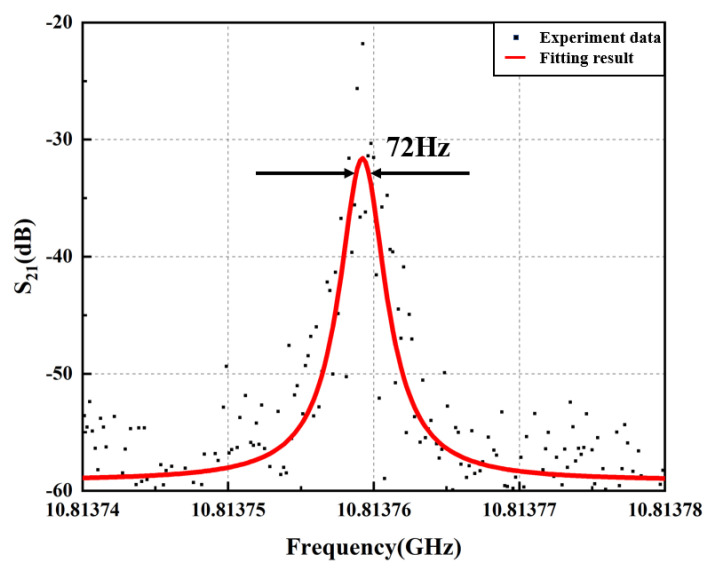
The amplified ultra narrow frequency response of MPF.

**Figure 6 micromachines-14-01322-f006:**
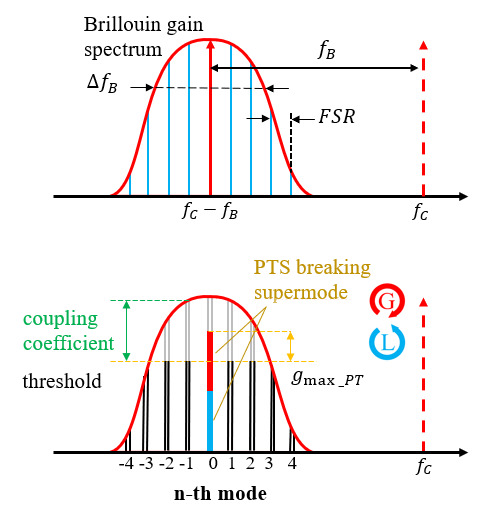
The principle and mechanism of PT symmetric BFL. (**a**) The process of SBS in spectrum. (**b**) Mode selection mechanism under PT symmetry broken condition in PT symmetric BFL.

**Figure 7 micromachines-14-01322-f007:**
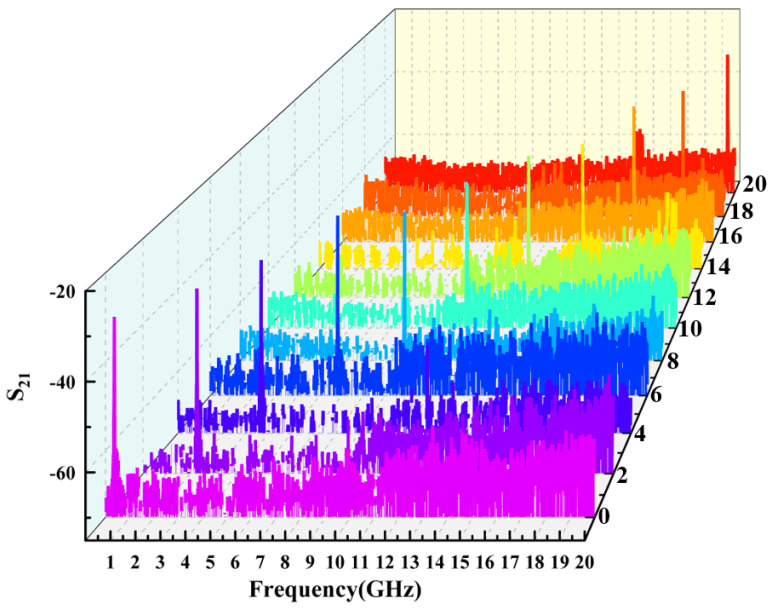
Frequency response of MPF corresponding to pump light wavelength range from 1549.9200 nm to 1550.0800 nm.

**Figure 8 micromachines-14-01322-f008:**
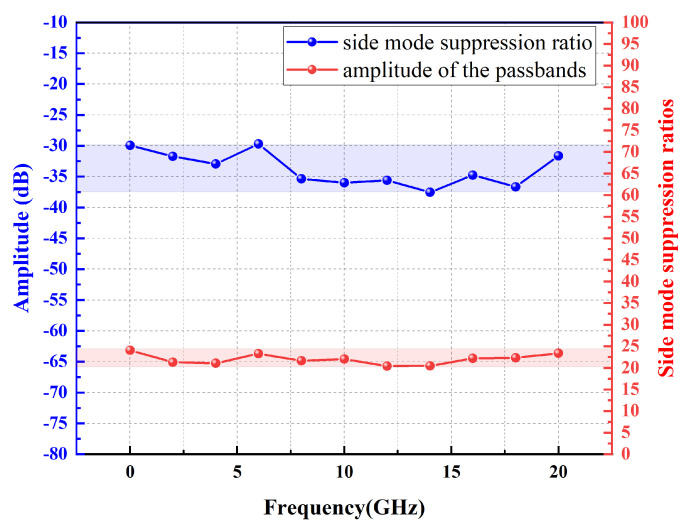
Out of band suppression ratio and frequency response amplitude test results of the 0–20 GHz internal filtering passband.

**Table 1 micromachines-14-01322-t001:** Performance comparison of various SBS based MPF technologies.

Technology	−3 dB Bandwidth (Hz)	Frequency Tuning Range (GHz)
MPF based on PTS BFL (This work)	72	0–20
MPF based on DR-BFL [23]	114	0–20
MPF cascaded active delay loop fiber recirculating [19]	150 k	0–40
MPF based on OEFL [21]	300 k	1–17
MPF cascaded fiber ring resonator [24]	825 k	2–16
MPF based on tailoring Brillouin gain and loss spectrums [20]	7.8 M	-
MPF cascaded DPMZMs [16]	55 M	0–9.6

Notes: PTS: Parity Time Symmetry, BFL: Brillouin fiber laser, OEFL: optical-electrical feedback loop, DPMZM: dual-parallel Mach–Zehnder modulators.

## Data Availability

The data that support the findings of this study are available upon reasonable request from the authors.

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
