# Peer review of "Ultra-Narrow Bandwidth Microwave Photonic Filter Implemented by Single Longitudinal Mode Parity Time Symmetry Brillouin Fiber Laser"

_micromachines, 2023, doi:10.3390/mi14071322_

Round 1

Reviewer 1 Report

The authors introduced the concept of parity-time symmetry mode selection into the microwave photonic filter to compress the bandwidth to sub-kHz. I think this work is worthy to be published in Micromachines, while I have the following comments to be responded from the authors.

1)      Please check the syntax of sentence reported at lines 87-90.

2)      “divice” should be “device” at line 102. Besides, please check the syntax of this sentence.

3)      How and why does EOM limit the adjustable range of MPF as stated at line 103-106.

4)      I recommend the expression of “release of FSR matching” replacing “not being affected by FSR matching difficuties” at line 111-112.

5)      The concept of PT is attractive recently, while the necessity and value of the introduction of PT into the MPF is unclear though the authors claimed some in the Introduction.

6)      Some typos should be corrected.

7)      Can the authors evaluate the efficiency of the system? Since so many couplers and circulars are involved, I do not expect a high efficiency.

Some typos and syntax should be checked and corrected throughout the manuscript.

Author Response

请参阅附件。

Reviewer 2 Report

The Authors propose a novel microwave photonic filter absed on single longittudinal mode Brillouin laser achieved by parity time symmetry mode selection. The results have been experimentally achieved. The manuscript is well written and it reports a significant improvement of the state-of-the-art and it deserves the publication after addressing the following minor comments.

Here, my comments:

1. In the Section 1, the literature should be enlarged by taking into account also integrated technologies (see, e.g.,  A monolithic integrated photonic microwave filter. Nature Photonics11(2), 124-129, 2017; Ultra-compact tuneable notch filter using silicon photonic crystal ring resonator. Journal of Lightwave Technology37(13), 2970-2980, 2019;  Low-power, chip-based stimulated Brillouin scattering microwave photonic filter with ultrahigh selectivity. Optica2(2), 76-83, 2015).
